# Nascent ribosomal RNA act as surfactant that suppresses growth of fibrillar centers in nucleolus

Tetsuya Yamamoto [1,2,5✉], Tomohiro Yamazaki [3,5✉], Kensuke Ninomiya[3] & Tetsuro Hirose [3,4]

Liquid-liquid phase separation (LLPS) has been thought to be the biophysical principle governing the assembly of the multiphase structures of nucleoli, the site of ribosomal biogenesis. Condensates assembled through LLPS increase their sizes to minimize the surface energy as far as their components are available. However, multiple microphases, fibrillar centers (FCs), dispersed in a nucleolus are stable and their sizes do not grow unless the transcription of pre-ribosomal RNA (pre-rRNA) is inhibited. To understand the mechanism of the suppression of the FC growth, we here construct a minimal theoretical model by taking into account nascent pre-rRNAs tethered to FC surfaces by RNA polymerase I. The prediction of this theory was supported by our experiments that quantitatively measure the dependence of the size of FCs on the transcription level. This work sheds light on the role of nascent RNAs in controlling the size of nuclear bodies.

[1] Institute for Chemical Reaction Design and Discovery, Hokkaido University, Kita 21, Nishi 10, Kita-ku, Sapporo 001-0021, Japan. [2] PRESTO, Japan Science and Technology Agency (JST), 4-1-8, Honcho, Kawaguchi, Saitama 332-0012, Japan. [3] Graduate School of Frontier Biosciences, Osaka University, 1-3 Yamadaoka, Suita 565-0871, Japan. [4] Institute for Open and Transdisciplinary Research Initiatives, Osaka University, 1-3 Yamadaoka, Suita 565-0871, Japan. [5] These authors contributed equally: Tetsuya Yamamoto, Tomohiro Yamazaki. ✉email: tyamamoto@icredd.hokudai.ac.jp; tyamazaki@fbs.osaka-u.ac.jp

In the interchromatin spaces of a cell nucleus, there are a variety of nuclear bodies, such as nucleoli[1–4], Cajal bodies[5,6], nuclear speckles[7], and paraspeckles[8]. Many of the nuclear bodies are scaffolded by ribonucleoprotein (RNP) complexes, which are composed of RNA and RNA-binding proteins (RBPs). The class of RNAs that are essential to the assembly of specific subcellular bodies, is called architectural RNA (arcRNA)[8–11]. A growing number of researches suggest that nuclear bodies are assembled via liquid-liquid phase separation (LLPS), which is driven by the multivalent interaction between RBPs bound to arcRNAs[8–11]. Condensates produced by LLPS are spherical and increase their size by coarsening and/or coalescence to minimize the surface energy (macroscopic phase separation)[12].

Nucleoli are nuclear bodies, where ribosome assembly takes place[1,2]. In the nucleoli, pre-ribosomal RNAs (pre-rRNAs) are transcribed by RNA polymerase I (Pol I) and maturated to construct ribosomes with ribosomal proteins. A nucleolus is not a uniform disordered liquid, but multiple microphases, called fibrillar centers (FCs), are dispersed in the sea of a granular component (GC) (Fig. 1a). Fibrillarin molecules (FBLs), which are RBPs interacting with pre-rRNAs, are condensed to form another phase, called the dense fibrillar component (DFC), at the interfaces between FCs and the GC. The multiphase structures of nucleoli have been thought to be assembled via simple LLPS[13]. However, this picture may not be complete because the FCs do not show coalescence or coarsening to increase their sizes. Indeed, when the Pol I transcription of pre-rRNA is inhibited, FCs show coalescence, as in the case of LLPS, and are excluded to the surface of the nucleolus (the excluded FCs are called nucleolar caps)[14]. This implies that transcription somehow suppresses the growth of FCs. Ribosomal DNA (rDNA), from which pre-rRNAs are transcribed, is a repeat sequence of coding units (10 kb) intervened by the intergenic regions (30 kb). The transcriptionally active rDNA units and Pol I are localized at the surfaces of FCs where the transcription of pre-rRNAs takes place[3,14–16]. The DFC layer is probably assembled by the RNP complexes of nascent pre-rRNAs and the associated RBPs, such as FBLs[16]. FBLs show phase separation with the physiological concentration and bind to nascent pre-rRNAs[13,16]. These experimental results imply that nascent pre-rRNAs may act as 'surfactants' that suppress the growth of FCs.

Paraspeckles are nuclear bodies scaffolded by NEAT1_2 arcRNAs. Analogous to FCs in the nucleolus, multiple paraspeckles are dispersed in a nucleoplasm[17]. Paraspeckles form the core-shell structure, where the two terminal regions of NEAT1_2 form the shell and the middle region of NEAT1_2 forms the core[18,19]. We have recently shown that the terminal regions of NEAT1_2 in the shell suppress the growth of paraspeckles, analogous to micelles of triblock copolymers[17,20]. The size of paraspeckles increases with the NEAT1_2 transcription upregulation[17,21], which is distinct from the response of FCs in a nucleolus to transcription upregulation. This implies that the size control mechanism of FCs in a nucleolus is different from that of paraspeckles.

We here construct a simple theoretical model that predicts the contributions of nascent pre-rRNAs to the assembly of the multiphase structure of nucleolus. This model takes into account the phase separation, the transcription kinetics, and the interfacial effects, which are probably the minimum to account for the multiphase structure. Our theory predicts that the nascent pre-rRNAs are stretched to accommodate FBLs in the DFC layer and generate the lateral pressure that counteracts the interfacial tensions. The size of FCs is determined by the balance of the interfacial tensions and the lateral pressure. The latter activity of nascent pre-rRNAs is analogous to the surface activity of surfactants. Our theory quantitatively predicts the dependence of the size of FCs on the transcription rate. The suppression of the FC growth by the pre-rRNA transcription results from the fact that these complexes are end-grafted to the FC surfaces via Pol I, in contrast to other condensates in which arcRNAs diffuse freely and increase their sizes by the transcription upregulation. To test our prediction, we experimentally measured the FC sizes and the pre-rRNA levels while the transcription rate is changed by the dose of BMH-21 or CX-5461, which specifically inhibits the Pol I transcription. The scaling exponent predicted by our theory is consistent with our experimental results, implying that the lateral pressure generated by nascent pre-rRNAs are possible mechanism of the size control of FCs. We anticipate that this theory can be a base theory to further look into the contribution of the multiphase structure in the function of nucleoli and can be extended to study the mechanism of the size control and the functions of other nuclear bodies, such as nuclear speckles and transcriptional condensates.

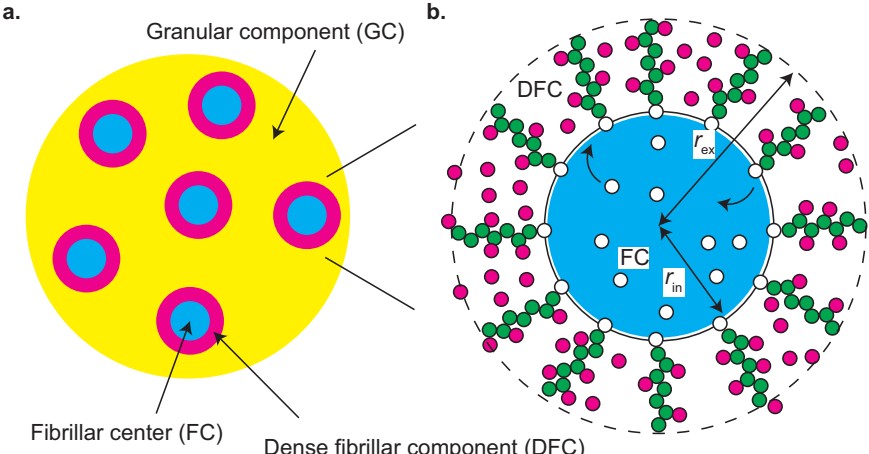

**Fig. 1 Multiphase structure of a nucleolus. a** A nucleolus is composed of multiple fibrillar centers (FC) microphases in the sea of the granular component (GC). There is a layer of dense fibrillar component (DFC) between each FC and GC. **b** RNA polymerase I (Pol I) molecules (white particles) are entrapped in FC microphases (light blue) and the active rDNA units (black line) are localized at the surfaces of microphases. Nascent pre-rRNAs (green particles) are thus at the surfaces of the microphase and form a DFC layer with RNA-binding proteins (magenta particles). The interface between FC and DFC is located at a distance $r_{in}$ from the center and the interface between DFC and GC is located at a distance $r_{ex}$.

## Results

**Model of nucleolus.** We here construct a minimal model of a nucleolus to predict the size of FCs in the steady state (the list of symbols is given in Supplementary Table 1). The nucleolus is composed of a GC in which multiple spherical FCs are dispersed (Fig. 1a). Pol I molecules (shown by white particles in Fig. 1b) are entrapped in the FCs (shown by a light blue droplet in Fig. 1b). The transcriptionally active repeat units of rDNA (shown by the black line in Fig. 1b) are localized at the surfaces of the FCs. We assume that the number of Pol I molecules and the copy number of transcriptionally active rDNA units in the nucleolus are constant. Nascent pre-rRNAs (shown by chains of green particles in Fig. 1b) are localized at the surfaces of FCs and form a DFC layer (Fig. 1b). Recent experiment revealed that the 5′ terminal external transcribed spacer (ETS) regions of nascent pre-rRNAs are localized at the DFC layer and that, among the proteins localized in the DFC layer, FBLs contributed most significantly to the localization of the terminal regions of nascent pre-rRNAs in the DFC layer[16]. Motivated by this result, we take into account only the FBL-binding terminal regions of nascent pre-rRNAs as arcRNAs that scaffold the DFC layers and only FBLs as RBPs that bind to these regions (see also the "Discussion" section). Recent experiments also revealed that the upstream half of ETSs are localized at the DFC layer and the downstream half is localized at the FC[16]. This motivated us to assume that the terminal regions are spanning from the top to the bottom of the DFC layer. Some RBPs bind to the nascent pre-rRNAs and the others diffuse freely in the DFC layer (shown by magenta particles in Fig. 1b). The interface between FC and DFC is located at a distance $r_{in}$ from the center and the interface between DFC and GC is located at a distance $r_{ex}$ from the center (see Fig. 1b). For simplicity, we assume that all the FCs have equal volume and the sum of the volumes of microphases is fixed to $V_m$. This assumption does not seem so far off although it has not been quantified[22].

The number of nascent pre-rRNAs at the surfaces of FCs is determined by the kinetics of transcription and RNA processing. We treat the transcription dynamics of pre-rRNA by using an extension of the model used by Stasevich and coworkers[23] (Fig. 2). Pol I in an FC binds to the transcription start site (TSS) of a transcriptionally active rDNA unit. The Pol I bound to the TSS starts transcription (Fig. 2a) or returns to the FC without starting transcription (Fig. 2a). The rate with which the Pol I bound to the TSS starts transcription is smaller than the rate with which this Pol I returns to the FC without starting transcription. The Pol I polymerizes a pre-rRNA while it moves uni-directionally towards the transcription termination site (TTS). The nascent pre-rRNAs are subject to co-transcriptional RNA processing (Fig. 2b). The FBL-binding terminal region is located at the ETS of pre-rRNA and is cleaved co-transcriptionally by endoribonuclease. The average time between the transcription initiation and the cleavage of the FBL-binding region is $\tau_{pr}$, where this time includes the elongation time to the cleavage site and the time necessary for the enzymatic reaction, see Fig. 2b and the Discussion. Pol I reaches TTS at the average elongation time $\tau_0$ and is then released to the FC ($\tau_{pr} < \tau_0$), see Fig. 2c. This model is designed to focus on the transcription of the FBL-binding terminal regions of nascent pre-rRNAs and to treat the rest of the process as simple as possible. The kinetic equations of the Pol I transcription and co-transcriptional processing are derived by taking into account the above-mentioned processes, see Fig. 2 and Eqs. (5)–(7). By solving these kinetic equations for the steady state, we found that the surface density $\sigma_{in}$ is proportional to the radius $r_{in}$ of FCs, see Eq. (9) in Methods. We thus introduce a dimensionless parameter

$$\zeta = N_r \frac{\sigma_{in} b^3}{r_{in}}, \tag{1}$$

which is proportional to the transcription rate and processing time $\tau_{pr}$, while it is independent of the radius $r_{in}$. $N_r$ is the number of units in the terminal region of a nascent pre-rRNA ($\approx 33$)[16] and $b$ is the (Kuhn) length of RNA unit ($\approx 4$ nm)[24].

The stability of the system is quantified by the free energy. The free energy $F_d$ of a DFC layer has 4 contributions: (1) the elastic free energy $f_{ela}$ of nascent pre-rRNA, (2) the mixing free energy $f_{mix}$ of RBPs and solvent, (3) the interaction free energy $f_{int}$ between RBPs (both freely diffusing in the DFC layer and bound to nascent pre-rRNA), and (4) the binding free energy $f_{bnd}$ of RBPs to the pre-rRNA terminal regions. The expressions of these free-energy contributions are shown in Eqs. (11)–(16) in Methods. The elastic free energy $f_{ela}$ of flexible polymer chains, such as nascent pre-rRNAs, increases as the chains stretch because the number of possible conformations decreases[12,25]. The thermal fluctuation mixes the components (nascent pre-rRNA units, RBPs, and solvent) and this contribution is quantified as the mixing free energy $f_{mix}$. The interaction free energy $f_{int}$ is the free-energy contribution of the multivalent interactions between RBPs. The binding free energy $f_{bnd}$ represents the free-energy gain due to the binding of RBPs to nascent pre-rRNA units and the thermal fluctuations that dissociate RBPs from nascent pre-rRNA units. The free energy $F_d$ takes into account the binding of RBPs to the terminal regions of nascent pre-rRNAs and the spherical geometry of the system in an extension of the Alexander model of polymer brush[26].

The free energy $F$ of the system has the form

$$\frac{F}{V_m} = \frac{3}{4\pi r_{in}^3} \left[ F_d + 4\pi r_{in}^2 \gamma_{in} + 4\pi r_{ex}^2 \gamma_{ex} \right], \tag{2}$$

where $\gamma_{in}$ is the interfacial energy per unit area (interfacial tension) at the interface between FC and DFC ($r = r_{in}$) and $\gamma_{ex}$ is the surface energy per unit area (interfacial tension) at the interface between DFC and GC ($r = r_{ex}$). The external radius $r_{ex}$ is determined by the conservation of the number of RNA units in the DFC layer in the steady state, see Eq. (10) in Methods. The interfacial tension results from the interactions between the components of interfacing phases. We therefore assume that the surface tensions, $\gamma_{in}$ and $\gamma_{ex}$, are proportional the local volume fraction of RBPs at the interfaces, with the proportional coefficient $\gamma_p$, see Eqs. (17) and (18) in Methods. The free energy $F_d$ is a functional of the occupancy $\alpha_p$ of the pre-rRNA terminal regions by RBPs, the volume fraction $\phi_p$ of RBPs freely diffusing in the DFC layer, and the volume fraction $\phi_r$ of the FBL units, where these are functions of the distance $r$ from the center of the FC. We derive these quantities by analyzing the condition of the minimum of the free energy, Supplementary Note 1. The relaxation dynamics of the terminal region of nascent pre-rRNAs is estimated to be faster than the elongation and thus its contribution is negligible, see Supplementary Note 2 and Supplementary Table 2.

**Parameter estimate suggests that the terminal regions of nascent pre-rRNAs are highly stretched.** The independent parameters involved in our theory and their estimates are summarized in Table 1. The kinetic constants involved in the Pol I transcription are included in $\zeta$, see Eq. (1). The energy gain due to the binding of RBPs to the pre-rRNA terminal regions is represented as $\epsilon k_B T$. We represent the magnitudes of the multivalent interaction between RBPs by using the Flory interaction parameter $\chi$. In principle, these parameters, $\epsilon$ and $\chi$, do not change with the suppression of Pol I transcription, but by the mutation of the RNA-binding regions and the intrinsically disordered regions of FBL, respectively. We neglect the dependence of the magnitudes of the interaction between RBPs on their binding state and also

**a.** Promoter binding/transcription start    **b.** RNA processing

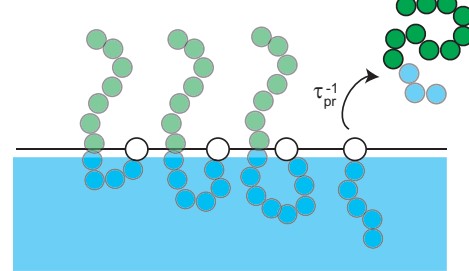

**c.** Termination

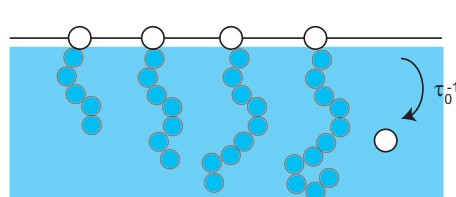

**a.** Pol I occupancy at TSS

$$\frac{dn_{\mathrm{ps}}}{dt} = k_{\mathrm{on}}\rho(1 - n_{\mathrm{ps}}) - k_{\mathrm{off}}n_{\mathrm{ps}} - k_{\mathrm{e}}n_{\mathrm{ps}}$$

**b.** Pol I with nascent pre-rRNA terminal region

$$\frac{dn_{\mathrm{pe}}}{dt} = k_{\mathrm{e}}n_{\mathrm{ps}} - \frac{n_{\mathrm{pe}}}{\tau_{\mathrm{pr}}}$$

**c.** Pol I without nascent pre-rRNA terminal region

$$\frac{dn_{\mathrm{p0}}}{dt} = \frac{n_{\mathrm{pe}}}{\tau_{\mathrm{pr}}} - \frac{n_{\mathrm{p0}}}{\tau_0}$$

**Fig. 2 Model of transcription dynamics.** DNA (black solid line) is localized at the surface of an FC (cyan). **a** RNA polymerase I (Pol I) in a microphase binds to the transcription starting site (TSS) of an active rDNA repeat unit. The bound Pol I starts transcription with the rate $k_{\mathrm{e}}$ or returns to the microphase without starting transcription. **b** During the transcription, Pol I migrates uni-directionally towards the transcription terminating site (TTS) while polymerizing a nascent pre-rRNA. The terminal region of the nascent pre-ribosomal RNA (pre-rRNA), to which FBL binds, are cleaved by the co-transcriptional RNA processing with the rate $\tau_{\mathrm{pr}}^{-1}$. The terminal region is released to GC. **c** After the cleavage of the terminal region, Pol I continues transcription until it reaches the TTS. At the TTS, Pol I is releaed to the FC with the rate $\tau_0^{-1}$.

**Table 1 List of independent (dimensionless) parameters involved in our theory.**

| Parameter | Meaning | Estimate | Value | Ref |
|---|---|---|---|---|
| $\zeta$ | Rescaled pre-rRNA surface density | <0.12 | Varied | 16 |
| $\frac{\gamma_p b^2}{N_r k_B T}$ | Rescaled surface tension per RBP | 0.03 | 0.03 | |
| $\chi$ | Interaction parameter | >10 | 12 | |
| $\epsilon$ | RBP-RNA-binding energy | <−10 | −12 | |
| $\mu_{\mathrm{p}}/(k_B T)$ | RBP chemical potential | −10 | −10 | 13 |

The unit length $b$ of pre-rRNA is estimated as 4 nm ($\approx$12 b) by using the value of single-stranded DNA (23). The absolute temperature $T$ is 300 K. The surface density $\sigma_{\mathrm{in}}$ was estimated by the number of Pol I (50 Pol I per rDNA, 300 copies of active rDNA repeat per cell, ~10 FCs per nucleolus, ~2 nucleoli per cell) and the typical radius of FCs $r_{\mathrm{in}} \sim$ 100nm (3,16). The surface tension $\gamma_p$ can be estimated as $\sim k_B T/b_p^2$ (27,28), where the size $b_p$ of FBL is $\approx$4 nm (29). The number of units $N_r$ in the FBL-binding region of pre-rRNA is estimated by using its length ~400 b (2,16). The chemical potential of FBL was estimated by using its concentration ~1 μM (13).

assume that solvent molecules and naked pre-rRNA units are equivalent in terms of the magnitudes of the interaction. This highlights the fact that the multivalent interaction between FBLs drives phase separation in vitro, not to lose the essence by introducing many interaction parameters. The volume fraction $\phi_{\mathrm{p0}}$ of RBPs in the nucleoplasm is taken into account via the chemical potential $\mu_{\mathrm{p}} = k_B T \log \phi_{\mathrm{p0}}$.

The interaction parameter $\chi$ and the binding energy $\epsilon$ are important quantities of polymer physics but are not experimentally well characterized. In the limit of vanishing volume fraction of nascent RNAs, $\sigma_{\mathrm{in}} \to 0$, our model returns to the theory of the binary mixture of RBPs and solvent molecules[12,27], see Eqs. (13)–(16) in Methods. The latter theory predicts that the phase separation happens at the threshold interaction parameter $\chi_{\mathrm{tr0}}(= -\mu_{\mathrm{p}}/(k_B T))$. FBL molecules show phase separation in vitro with the physiological concentration[13], implying that $\chi > \chi_{\mathrm{tr0}}$. If FBLs bind to the pre-rRNA terminal regions even in a dilute solution, it implies that $\epsilon < \mu_{\mathrm{p}}/(k_B T)$. The size of the FBL-binding terminal region of a nascent pre-rRNA is $\approx$30 nm in the relaxed state

($\approx$130 nm in the maximally stretched state), while the thickness of a DFC layer is $\approx$100 nm, implying that FBLs are highly stretched.

**The multivalent interaction between FBLs localizes these RBPs to the DFC layer and stretches the terminal regions of nascent pre-rRNAs.** To avoid complexity arising from the geometry of the system, we first analyze the composition of a DFC layer on a planer surface. This is indeed the limit of large FCs and the essential feature of the assembly of DFC layers is included in this analysis. In the planer geometry, the chemical potential of RBPs and the osmotic pressure are uniform, see also the asymptotic analysis in Supplementary Note 3.

The magnitude of the multivalent interactions between RBPs (either bound to the pre-rRNA terminal regions or freely diffusing) increases with increasing the interaction parameter $\chi$. Although it is not straightforward to experimentally control the interaction parameter $\chi$ of FBLs, it is instructive to analyze the composition of the layer occupied by the FBLs of nascent pre-rRNAs as a function of the interaction parameter $\chi$, see Fig. 3. For cases in

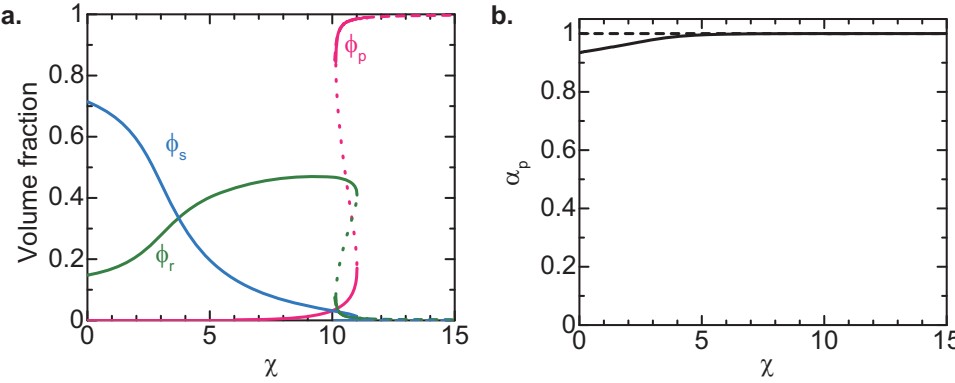

**Fig. 3 Composition of planer DFC layer vs. interaction parameter $\chi$. a** The volume fraction $\phi_r$ of pre-rRNA units (green), the volume fraction $\phi_p$ of freely diffusing RBPs (magenta), the volume fraction $\phi_s$ of solvent molecules (cyan) in a DFC layer are shown as functions of the interaction parameter $\chi$. **b** The occupancy $\alpha_p$ of pre-rRNA terminal region by RBPs is shown as a function of the interaction parameter $\chi$. We used $\mu_p/(k_B T) = -10.0$, $\epsilon = -12.0$, $\sigma_{in} b^2 = 0.05$, and $\Pi_{ex} b^3/(k_B T) = 0.0$ for the calculations, see also Table 1. The solid lines are derived by numerically solving Supplementary Equations (S15), (S16), and (S20). The broken lines are derived by using Supplementary Equation (S47) (shown for $\chi > 10.1$).

which the interaction parameter $\chi$ is small (which corresponds to self-association defective FBL ΔIDR mutant[16]), the major component of the layer is solvent (nucleoplasm), see the cyan line in Fig. 3. For $\chi < \chi_s$, the excluded volume interaction between nascent pre-rRNA units dominates the attractive interaction between RBPs bound to these nascent pre-rRNAs (good solvent regime). The interaction parameter $\chi_s$ at the crossover is 2 in the limit of $\epsilon \ll \mu_p/(k_B T)$, such as the case of Fig. 3, see also Supplementary Equation (S27) for the general expression. The inverse of the volume fraction $\phi_r$ of nascent pre-rRNA is proportional to the extent of the stretching of the terminal regions of nascent pre-rRNAs. The terminal regions of these RNAs are somewhat stretched for $\chi \approx 0$ due to the excluded volume interaction between the pre-rRNA units. The terminal regions shrink as the magnitude $\chi$ of the attractive interaction between RBPs bound to the terminal regions of nascent pre-rRNAs, see the green line in Fig. 3. For $\chi_s < \chi < -\mu_p/(k_B T)$, the attractive interaction dominates the excluded volume interaction (poor solvent regime) and the terminal regions are collapsed with RBPs bound to these regions (melt regime), see the green line in Fig. 3.

At a threshold interaction parameter, $\chi_{th} \approx -\mu_p/(k_B T)$, the volume fraction $\phi_p$ of freely diffusing RBPs jumps to a large value, see the magenta line in Fig. 3. In contrast, the volume fraction of the solvent (nucleosol) jumps to almost zero, see the cyan line in Fig. 3. The feature of the layer for $\chi > \chi_{th}$ is analogous to the DFC layer observed experimentally (which we thus call DFC regime). The volume fraction $\phi_r$ of nascent pre-rRNA units jumps to a small value at $\chi \approx \chi_{th}$, see the green line in Fig. 3, implying that the terminal regions of the nascent pre-rRNAs are stretched at the transition. Because the volume fraction $\phi_r$ is smaller than the case of $\chi = 0$, the stretching is not solely due to the fact that RBPs in the DFC layer behave as an athermal solvent to the RNP complexes of RBPs and the pre-rRNA terminal regions. The terminal regions stretch so that the DFC layer can accommodate more RBPs to decrease the free energy due to the multivalent interactions between them. The stretching thus may be a transport mechanism of the ETS of pre-rRNAs towards the DFC layer.

**The lateral pressure generated by nascent pre-rRNAs suppresses the growth of FCs.** Now we discuss the DFC layers at the surfaces of spherical FCs to derive the radius of FCs in the steady state. In the spherical geometry, the chemical potential of RBPs and the chemical potential of pre-rRNA units, instead of the osmotic pressure, are uniform. These are the conditions of the minimum of the free energy, Eq. (2), see also Supplementary Note 1. We performed numerical and analytical calculations for

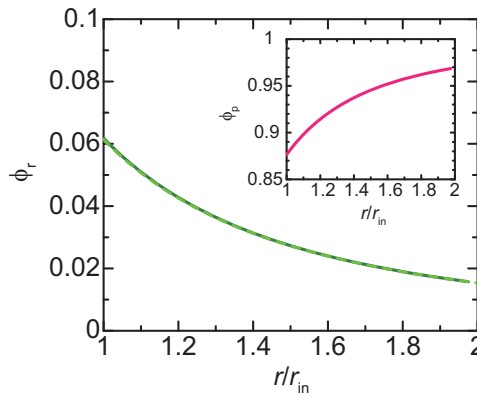

**Fig. 4 Profile of volume fraction $\phi_r$ of pre-rRNA units in DFC layer.** The volume fraction of nascent pre-rRNA units is shown as a function of the position $r$ in the DFC layer ($r_{in} < r < r_{ex}$). The solid dark green line is derived by numerically calculating Supplementary Equations (S13), (S15), and (S16) for $\zeta = 0.06$ with the condition that the volume fraction of solvent is zero and the occupancy of the terminal regions of pre-rRNAs by RBPs is unity. The values of other parameters are summarized in Table 1. The broken light green line is derived by using Supplementary Equation (S59) with $\phi_{ex} = 0.0157$ and $r_{ex} = 1.98$ (which were derived from the numerical calculation to obtain the light green line). The volume fraction of freely diffusing RBPs is shown in the inset.

the DFC regime in which the volume fraction of solvent molecules is negligibly small and the occupancy $\alpha_p$ of the pre-rRNA terminal regions is approximately unity, see also Supplementary Note 4. In the spherical geometry, the volume fraction $\phi_r$ of pre-rRNA units is not uniform in the DFC layer and decreases with the distance $r$ from the center ($r_{in} < r < r_{ex}$), see the solid green line in Fig. 4. It results from the fact that the elastic stress (proportional to the number of pre-rRNAs per unit area) generated by pre-rRNA decreases with increasing $r$. The rest of the space in the DFC layer is filled by RBPs, see the inset of Fig. 4.

We used the profile of the pre-rRNA units to derive the free energy as a function of the radius $r_{in}$ of FCs. For cases in which the Pol I transcription is inhibited, the free energy of the system decreases monotonically with the radius $r_{in}$, implying that, the radius of FCs increases with time by coarsening or coalescence. For cases in which the Pol I transcription is active, the free energy generated by the terminal regions of the nascent pre-rRNA is an intriguing function of $r_{in}$, which has a minimum, see the solid

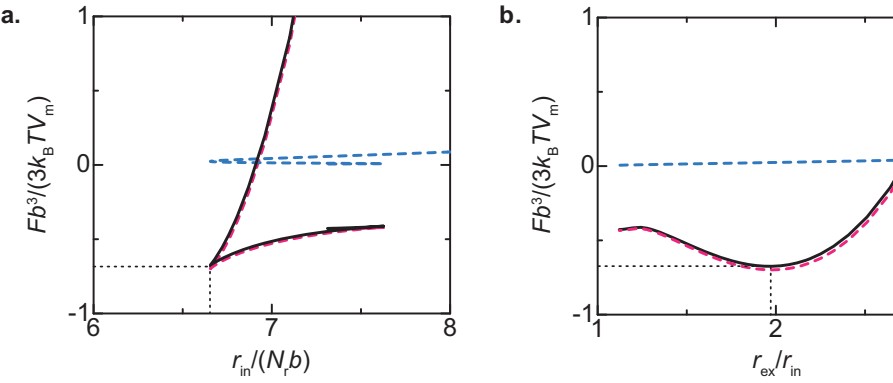

**Fig. 5 Free energy $F$ of nucleolus.** The free energy $F$ of the system is shown as a function of the radius $r_{in}$ of FCs (**a**) and the ratio $r_{ex}/r_{in}$ (**b**). The black solid line is the total free energy, including the free energy of DFC layers (shown by the magenta broken line, the first term of Eq. (2)) and the surface free energy (shown by the cyan broken line, the second and third terms of Eq. (2)) for $\zeta = 0.06$. The values of other parameters are summarized in Table 1.

black line in Fig. 5a. Indeed, the free energy is a simpler function of the ratio $r_{ex}/r_{in}$ with a minimum, see Fig. 5b. Indeed, the surface free energy, the second and third terms of Eq. (2), is not significant for the values of parameters estimated in Table 1, see the broken cyan line in Fig. 5a.

One of the conditions satisfied at the minimum of the free energy has an approximate expression

$$\Delta\Pi_{\parallel} \approx \gamma_{in} + \gamma_{ex}\frac{r_{ex}^2}{r_{in}^2}. \tag{3}$$

Indeed, Eq. (3) represents the balance of the surface tensions, $\gamma_{in}$ and $\gamma_{ex}$, and the lateral osmotic pressure $\Delta\Pi_{\parallel}$ generated in the DFC layer, see Eq. (20) for the complete form. The anisotropy in the osmotic pressure $\Delta\Pi_{\parallel}$ results from the fact that the elastic stress of pre-rRNAs only acts in the radial direction. Although the isotropic osmotic pressure generated by FBLs filled in the DFC layer is balanced by the elastic stress of pre-rRNAs in the normal direction, it still acts in the lateral direction. The radius $r_{in}$ at the minimum of the free energy decreases with increasing the transcription rate, see Fig. 6. The radius $r_{in}$ at the minimum of the free energy has an asymptotic form

$$\frac{r_{in}}{bN_r} = \frac{2}{\sqrt{3}}\left(\chi + \frac{\mu_p}{k_BT}\right)^{\frac{1}{2}}\zeta^{-\frac{1}{2}}, \tag{4}$$

see also the broken line in Fig. 6. Equation (4) is derived by expanding the free energy in a power series of $\phi_r$ and by using the asymptotic form of $\phi_r$, see Supplementary Note 4 for the details of the derivation. We also neglected the surface free energy, the second and third terms of Eq. (2), because it is not significant with our estimate in Table 1. Our theory therefore predicts that the radius of FCs decreases with the inverse of the square root of the transcription rate. At the minimum of the free energy, the ratio of the radii is $\frac{r_{ex}}{r_{in}} \approx 2$, which roughly agrees with the fact that the radius of FCs and the thickness of DFC layers are the same order of length scales, $\propto 100$nm, see Fig. 5b.

**Mild inhibition of Pol I increases the size of FCs**. The main prediction from our theoretical model is that the size of FCs increases by reducing the expression level of nascent pre-rRNAs. To experimentally test this prediction, we used BMH-21 and CX-5461, specific Pol I inhibitors, which reduce nascent pre-rRNA levels in a dose-dependent manner[28–30], see Figs. S1a, b. In untreated cells, small foci of UBF (a marker for FCs) and FBL (a marker for DFC) proteins were dispersed within the nucleoli as previously reported[14], see Fig. 7a and Figs. S2a, S2b, and S3a. In

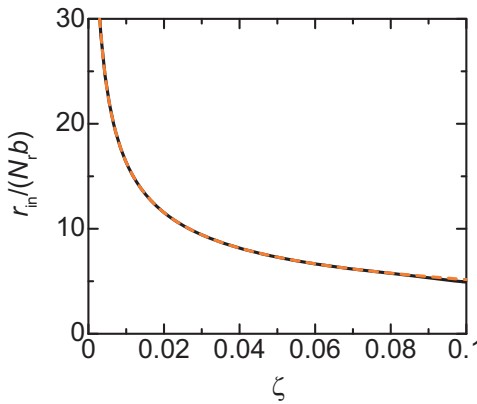

**Fig. 6 Radius $r_{in}$ of FCs vs. rescaled transcription rate $\zeta$.** The radius $r_{in}$ of FCs at the free-energy minimum is shown as a function of rescaled transcription rate $\zeta$. The solid dark green line is derived by numerically calculating Supplementary equations (S13), (S15), and (S16) with the condition that the volume fraction of solvent is zero and the occupancy of the terminal regions of pre-rRNAs by RBPs is unity. The orange broken line is derived by using Eq. 5. The parameters used for the calculations are summarized in Table 1.

contrast, UBF and FBL proteins were relocalized to nucleolar caps in the presence of high doses of BMH-21 (0.5 μM) or CX-5461 (2 μM), as reported[14,28], see Fig. 7a and Figs. S2a, S2b, and S3a. Strikingly, the FCs were larger in cells treated with the medium doses of BMH-21 (0.0625, 0.125, and 0.25 μM) or CX-5461 (0.25, 0.5, and 1 μM) than in untreated cells, Fig. 7a and Figs. S2a, S2b, and S3a. The range of concentrations of BMH-21 used in our experiments is well within the subsaturation regime of Pol I transcription[29]. We then quantified the size of the FCs under these untreated and mildly treated conditions, see Fig. 7b, c, and Figs. S3b, S3c, and S4. Indeed, the longest axis (Lx) and area of the FCs increase with increasing the dose of BMH-21 and CX-5461. As theoretically predicted, the Lx and area of the FCs increase according to pre-rRNA level reduction, see Fig. 7d, e and Figs. S3d, e.

**FC radius is a power function of the pre-rRNA transcription level with an exponent of approximately –0.5**. The scaling exponent $-1/2$ that describes the dependence of the size of FCs on the transcription level is a universal quantity that does not depend on specific values of the parameters, see Eq. (4). In

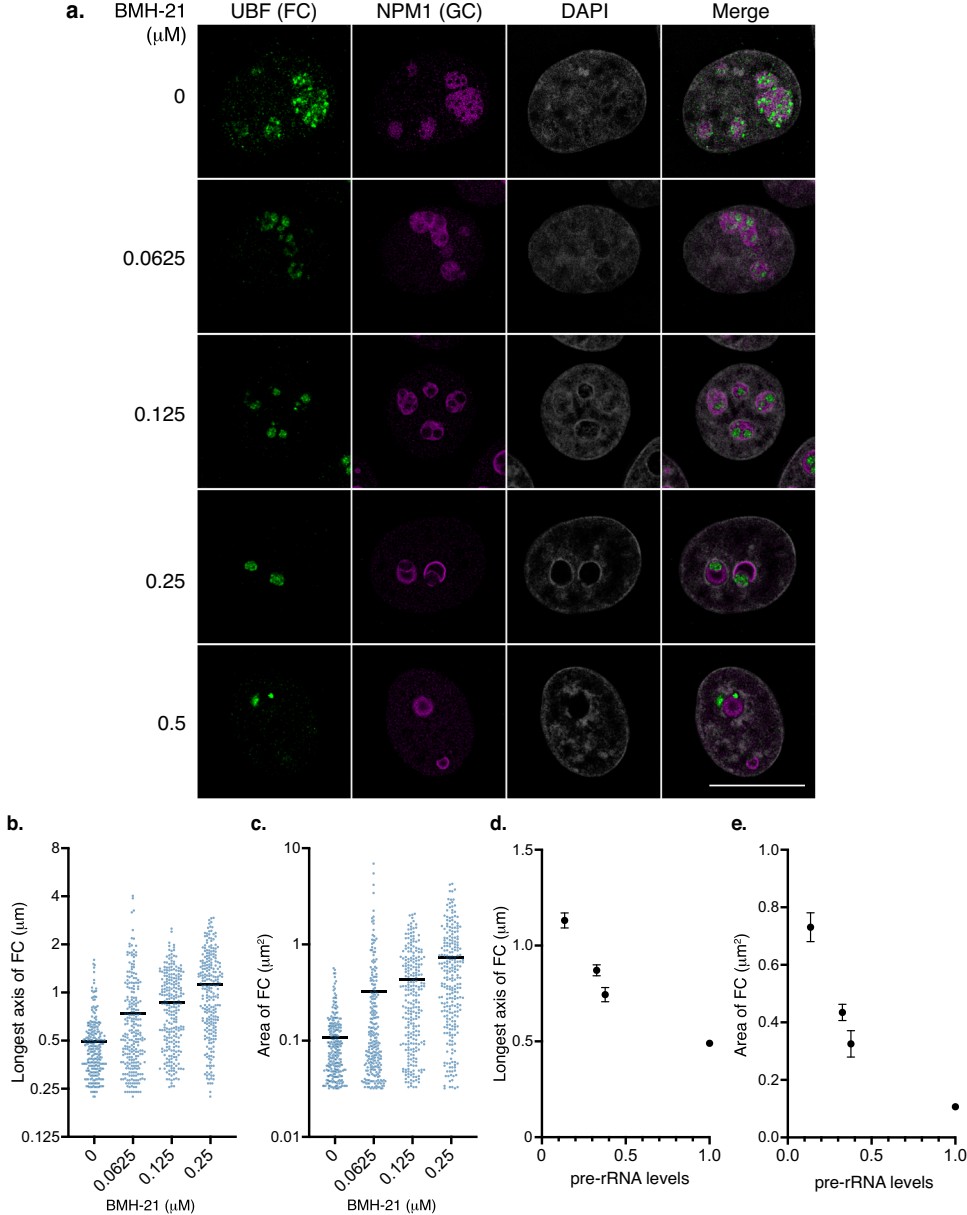

**Fig. 7 Mild Pol I inhibition by BMH-21 increases the size of FCs. a** Immunofluorescence of UBF (FC) and NPM1 (GC) in HeLa cells with or without BMH-21 treatments. Scale bar, 10 μm. **b**, **c** Quantification of the longest axis (**b**) and area (**c**) of the FCs in cells under indicated conditions. Each scatter dot plot shows the mean (black line). Dots indicate all points of quantified data ($n = 250$). Mean longest axes of the FCs is shown below: 0 μM: 0.4907 μm, 0.0625 μM: 0.7440 μm, 0.125 μM: 0.8710 μm, 0.25 μM: 1.131 μm. Mean areas of the FCs are shown below: 0 μM: 0.170 μm$^2$, 0.0625 μM: 0.3255 μm$^2$, 0.125 μM: 0.4347 μm$^2$, 0.25 μM: 0.7312 μm$^2$. Statistical analyses using the Kruskal–Wallis test with Dunn's multiple comparison test were performed and the results are shown as follows. **b** 0 μM vs. 0.0625 μM: $P < 0.0001$, 0 μM vs. 0.125 μM: $P < 0.0001$, 0 μM vs. 0.25 μM: $P < 0.0001$, 0.0625 μM vs. 0.125 μM: $P < 0.0001$, 0.0625 μM vs. 0.25 μM: $P < 0.0001$, 0.125 μM vs. 0.25 μM: $P = 0.0008$. **c** 0 μM vs. 0.0625 μM: $P = 0.0003$, 0 μM vs. 0.125 μM: $P < 0.0001$, 0 μM vs. 0.25 μM: $P < 0.0001$, 0.0625 μM vs. 0.125 μM: $P < 0.0001$, 0.0625 μM vs. 0.25 μM: $P < 0.0001$, 0.125 μM vs, 0.25 μM: $P = 0.0010$. **d** and **e** Graphs showing the mean longest axis (**d**) and area (**e**) of the FCs with SEM vs. pre-rRNA expression levels. The pre-rRNA expression level in untreated cells is defined as 1.

polymer physics, checking such scaling exponents with experiments is the first thing to do to test theories[31]. To extract the scaling exponent from our experiments, we curve-fit the size of FCs (the square root of the average area of FCs, which is proportional to their radii) as a function of the transcription level by a power function, see Fig. 8. By fitting the data with a power function, we found that the scaling exponent is –0.49 for the case of the BMH-21 treatment, –0.46 for the case of the CX-5461, and –0.48 if both data are fitted. The scaling exponent predicted by our theory –0.5 reasonably agrees with our experiments, see Eq. 4.

## Discussion
Our theory predicts that the terminal regions of nascent pre-rRNAs at the surfaces of FC microphases in a nucleolus generate the lateral osmotic pressure and suppress the growth of microphases. This theory predicts the dependence of the radius of FCs on the transcription rate is consistent with our experiments. The lateral pressure is generated by the fact that the nascent pre-rRNAs are tethered to the surfaces of FCs via Pol I and attract FBL to the DFC layer. This pressure increases with the transcription rate by increasing the surface density of the nascent pre-

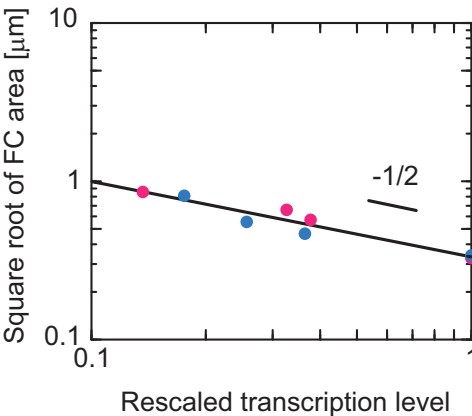

**Fig. 8 Exponent that accounts for the dependence of the size of FCs on pre-rRNA expression level.** The data in Fig. 7e was shown in the double-logarithm plot and fitted with a power function. The magenta and cyan dots are the results of suppressing the Pol I transcription by using BMH-21 and CX-5461. The slope of the double-logarithm plot is the exponent that accounts for the dependence of the radius of FCs on the transcription level. The curve fitting shows that the exponent is –0.49 for the case of the BMH-21 treatment, –0.46 for the case of the CX-5461, and –0.48 if both data are fitted.

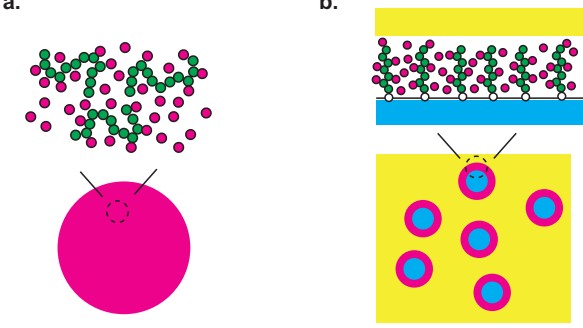

Condensate growth enhanced     Condensate growth suppressed

**Fig. 9 Summary of results.** RNP complexes enhance or suppress the growth of condensates depending on whether the RNP complexes are mobile in the interior or tethered to the surfaces of the condensates. **a** The multivalent interaction between the RNP complexes enhances the growth of the condensates if these condensates are assembled by the RNP complexes. **b** The multivalent interaction between the RNP complexes suppresses the growth of the condensates if these complexes are tethered to the surfaces of the condensates assembled by other RNAs and proteins.

rRNAs. Thus, if the Pol I transcription is continuously and slowly suppressed, the size of FCs increases following Eq. (4), see also Fig. 6. The contributions of the kinetics of transcription to the lateral pressure remain to be experimentally determined, while we estimate that its contribution is relatively small, see Supplementary Note 2. Our theory also predicts that the radius of FCs increases with increasing the pre-rRNA processing rate $\tau_{pr}^{-1}$. The inhibition of processing induced by ribosomal protein depletion, such as uL18 (RPL5) and uL5 (RPL11), changes the multiphase structure of nucleoli[32]. Recent experiments have demonstrated that long pre-rRNAs are accumulated in RPL5 knockdown cells and the size of FCs in such cells is smaller than that in the wild type[33]. The consistency between our theory and experiments suggests that the growth of FCs is suppressed by the terminal regions of nascent pre-rRNAs localized at the surfaces of FCs. This implies that the elastic energy of the terminal region is the driving force of the transport of the 5' ETSs cleaved by RNA processing towards GC.

In many cases, the multivalent interaction between complexes of RNA and RBPs drives the growth of condensates[34,35], see Fig. 9a. The size of such condensates increases with increasing the transcription rate[17,21,36]. In contrast, in the case of FCs in nucleoli, the multivalent interaction between the complexes of nascent RNA and RBPs rather suppresses the growth of FCs[14]. The size of FCs indeed decreases with increasing the transcription rate. Our theory predicts that the suppression of the growth of FCs by transcription results from the fact that pre-rRNAs are end-grafted to the surfaces of FCs via Pol I, see Fig. 9b.

Our theory predicts that the function of the terminal regions of nascent pre-rRNAs is analogous to surfactants, such as lipids. If the surfactants at an interface between two immiscible liquids are dense enough to form a monomolecular liquid film, these surfactants decrease the interfacial tension to almost zero to disperse multiple droplets even in the thermodynamic equilibrium[27,37]. The surfactant monolayer acts as a kinetic barrier to suppress the coalescence and decreases the hydrostatic pressure of the droplet to suppress the coarsening[12,27]. These features are analogous to the roles of nascent pre-rRNAs predicted by our theory. The surface activity of some RNAs and proteins was also discussed in synthetic systems[38,39]. There are also differences between nascent pre-rRNA and surfactants. Surfactants are localized at interfaces

at the thermodynamic equilibrium, while the terminal regions of nascent pre-rRNAs are localized at the FC-DFC interfaces only during the transcription, and their surface activity is regulated by Pol I transcription dynamics.

Although our theory probably could capture the essential features that are necessary to understand the assembly mechanism of the multiphase structure of nucleolus, it is not complete. First, the layers at the interface between FCs and GC are not uniform, but DFC regions are flanked by the regions, in which FBLs are not observed by super-resolution microscopy and are not visible by electron microscopy. Second, we did not take into account the downstream pre-rRNA regions that are not bound by FBLs. These downstream regions were observed in FCs[16]. The direct contribution of rDNAs localized at the surfaces of FCs is probably not significant, see Supplementary Discussion. Third, we did not explicitly take into account the transport dynamics of pre-rRNAs in GCs. Fourth, RNAs can be not only cleaved but also can be folded and/or chemically modified during co-transcriptional processing. The processing of pre-rRNAs in yeast has been well-documented[40,41], but much less in higher eukaryotes[42,43], in which the multiphase structure of nucleolus is usually observed. It is of interest to extend the present theory by taking into account the co-transcriptional RNA processing to study the biochemical role of RBPs in RNA processing[44], the modulation of RBP binding by the chemical modification of pre-rRNAs, such as methylation and pseudouridylation, and the role of the RNA chemical modification in the assembly of the multiphase structure of the nucleus. Fifth, we did not take into account the modulation of the stiffness of pre-rRNAs by the binding of RBPs. Sixth, we neglect the contributions of molecular factors other than the RNP complexes of nascent pre-rRNAs and RBPs. Last, but not least, we assume that the DFC layer is a liquid layer, see also Supplementary Discussion. Indeed, recent experiments showed that long non-coding RNA SLERT facilitates the transition of the RNA helicase DDX21 to the closed conformation and ensures the fluidity of proteins in the DFC layer[45].

Whether the multiphase structure of the nucleolus contributes to its function is not yet clear. It is of interest to extend our theory to understand the structure-function relationship of the nucleolus[46]. A peculiar process is that during the maturation of rRNAs, processed pre-rRNAs, from which ETSs are cleaved and thus have the affinity to FCs, penetrate through the DFC layer towards the GC.

One possible hypothesis is that the processed pre-rRNAs pass the flanking regions of the DFC layer. These flanking regions increase the density of the terminal regions of pre-rRNAs in the DFC compartments and enhance the lateral osmotic pressure, see also Supplementary Note 4. The enhanced lateral pressure and/or the kinetic trapping of the coalescence of FCs by nascent pre-rRNAs may explain the fact that the radius of FCs observed in experiments is somewhat smaller than that predicted by our theory. However, following the approach of classical theoretical physics, our present theory should be experimentally tested before adding more assumptions that have not been experimentally explored. Identifying RBPs bound to different regions of pre-rRNAs would be useful to address the above-mentioned questions.

Our theory may provide insight into the size control mechanism of other nuclear bodies. Nuclear stress bodies (nSBs) form a sea-island multiphase structure, which is somewhat analogous to nucleoli[47]. nSBs are composed of HSATIII arcRNA and specific RBPs[10]. HSATIII is transcribed by Pol II from the peri-centromeric Satellite III regions, which are enriched in tandem repeats[48,49]. The size of nuclear speckles increases by the suppression of transcription[7]. The size and stability of transcriptional condensate[50–54] also depend on the transcription rate and the length of transcripts[50,51]. These features are common with nucleoli and thus motivate us to think of a general mechanism involved in the assembly of the multiple subcompartments. The nascent RNAs, which are produced by transcription and are modulated by RNA processing, are important elements that regulate the multiphase structures and possibly their functions of nuclear bodies.

## Methods

**Transcription and RNA processing**. The occupancy $n_{ps}$ of the TSS of the active rDNA by Pol I follows the kinetic equation

$$\frac{d}{dt} n_{ps} = k_{on}\rho\left(1 - n_{ps}\right) - k_{off} n_{ps} - k_e n_{ps}. \tag{5}$$

Equation (5) suggests that the occupancy $n_{ps}$ changes due to the binding of Pol I from the FC (the first term), the unbinding of Pol I from the TSS to the FC without starting transcription (the second term), and the initiation of the elongation of Pol I (the third term), see Fig. 2a. $k_{on}$ is the rate constant for the binding of Pol I to TSS and $\rho$ is the concentration of Pol I in a FC (which will be determined later). $k_{off}$ is the rate constant for the unbinding of Pol I from TSS. $k_e$ is the rate of the transition of Pol I from the bound state to the elongation state.

The number $n_{pe}$ of elongating Pol I protein complexes with pre-processed nascent pre-rRNA follows the kinetic equation

$$\frac{d}{dt} n_{pe} = k_e n_{ps} - \frac{n_{pe}}{\tau_{pr}}. \tag{6}$$

Equation (6) suggests that the number $n_{pe}$ changes due to the release of Pol I from TSS for elongation (the first term) and the cleavage of the FBL-binding terminal region of nascent pre-rRNA (the second term), see Fig. 2b. $\tau_{pr}$ is the average time between transcription initiation and the cleavage of the terminal regions of the pre-rRNA.

The number $n_{p0}$ of elongating Pol I with processed nascent pre-rRNA follows the kinetic equation

$$\frac{d}{dt} n_{p0} = \frac{n_{pe}}{\tau_{pr}} - \frac{n_{p0}}{\tau_0}. \tag{7}$$

Equation (7) suggest that the number $n_{p0}$ changes due to the the cleavage of the terminal regions of pre-rRNAs (the first term) and the transcription termination (the second term), see Fig. 2c. $\tau_0$ is the average time between the cleavage of FBL-binding region and the transcription termination.

The numbers, $n_{ps}$, $n_{pe}$, and $n_{p0}$, of Pol I in each state are derived for the steady state, $\frac{dn_{ps}}{dt} = \frac{dn_{pe}}{dt} = \frac{dn_{p0}}{dt} = 0$, by using Eqs. (5)–(7). The inverse time $k_e$ of the transcription initiation is much smaller than the inverse $k_{off}$ of the unbinding of Pol I from TSS, $k_e \ll k_{off}$. The concentration $\rho$ Pol I in a FC is determined by the condition

$$(n_{ps} + n_{pe} + n_{p0})N_a + \rho V_m = N_{pol} \tag{8}$$

that the sum of the number of Pol I during transcription (the first term) and the number of Pol I diffusing in microphases (the second term) in the system is constant. All the parameters involved in Eq. (8) do not depend on the size of FCs, see Eqs. (5)–(7). The concentration $\rho$ of Pol I in FCs thus does not depend on the size of FCs.

The surface density $\sigma_{in}$ of the FBL-binding terminal regions of nascent pre-rRNAs at each FC (the number of nascent pre-rRNAs per unit area of each FC surface) is represented as

$$\sigma_{in} = \frac{n_{pe}}{4\pi r_{in}^2}\left(\frac{4\pi r_{in}^3}{3V_m}N_a\right) = \frac{1}{3}k_e\tau_{pr}\frac{\rho}{\rho + K_{pl}}\frac{N_a}{V_m}r_{in}. \tag{9}$$

by assuming that the terminal regions of nascent pre-rRNA is distributed uniformly at the surfaces of FCs. We note that $3V_m/(4\pi r_{in}^3)$ is the number of FCs in a nucleolus. The last form of Eq. (9) is derived by using the solution of Eqs. (5)–(7).

**Thickness of DFC layer**. The exterior radius $r_{ex}$ (see also Fig. 1b) is determined by the relationship

$$\int_{r_{in}}^{r_{ex}} dr 4\pi r^2 \phi_r = 4\pi r_{in}^2 \sigma_{in} N_r, \tag{10}$$

where $N_r$ is the average number of units in the terminal region of a nascent pre-rRNA. It is a mean-field treatment that assumes that nascent RNAs are composed of the same number of units and is effective within the Alexander-de Gennes approximation, with which the brush height is determined by the distance between neighboring grafting points and the average number of units per chain (if one neglects the fact that the lateral fluctuations of a chain composed of $N_r$ units is limited to $\sim N_r^{1/2}b$)[55,56].

**Free energy of DFC layer**. The free energy of each DFC layer has the form

$$F_d = \int_{r_{in}}^{r_{ex}} dr \frac{4\pi r^2}{b^3} f_d \tag{11}$$

with

$$f_d = f_{ela} + f_{mix} + f_{int} + f_{bnd} - \mu_p\left(\phi_p + \alpha_p\phi_r\right) + \Pi_{ex}b^3. \tag{12}$$

Equation (11) is the functional of the local volume fraction $\phi_r$ of nascent pre-rRNA units, the local volume fraction $\phi_p$ of RBPs, and the occupancy $\alpha_p$ of nascent pre-rRNA units by RBPs, which are functions of the distance $r$ from the center of the FC ($r_{in} < r < r_{ex}$), see Fig. 1b. $b$ is the length of a pre-rRNA unit. This free energy is composed of 4 contributions: $f_{ela}$ is the elastic free-energy density of the terminal regions of nascent RNAs, $f_{mix}$ is the free-energy density due to the mixing entropy of RBPs and solvent molecules, $f_{int}$ is the free-energy density due to the interactions between RBPs, and $f_{bnd}$ is the free-energy density due to the binding of RBPs to the terminal regions of nascent RNAs. $\mu_p$ is the chemical potential of RBPs and $\Pi_{ex}$ is the osmotic pressure from the exterior.

The (entropic) elastic free-energy density has the form

$$\frac{f_{ela}}{k_B T} = \frac{3}{2}\frac{b^4 \sigma^2(r)}{\phi_r}, \tag{13}$$

where $\sigma(r)$ is the surface density of FBL-binding regions of nascent pre-rRNAs. $\sigma(r) = \sigma_{\text{in}} r_{\text{in}}^2 / r^2$ for cases in which the surface of FCs is sphere and $\sigma(r) = \sigma_{\text{in}}$ for cases in which the surface of FCs is planer (which corresponds to the limit of $\frac{r_{\text{ex}} - r_{\text{in}}}{r_{\text{in}}} \ll 1$). The volume fraction $\phi_r$ of pre-rRNA units decreases as the terminal regions of the nascent pre-rRNAs are stretched. Equation (13) represents the fact that the elastic free energy $f_{\text{ela}}$ increases as these terminal regions are stretched. Equation (13) takes into account the spherical geometry of the system in an extension of the Alexander model (that assumes that the concentration of the terminal regions of nascent pre-rRNA is uniform in the DFC layer in the limit of $\frac{r_{\text{ex}} - r_{\text{in}}}{r_{\text{in}}} \ll 1$) in the spirit of the Daoud-Cotton theory[57], see Supplementary Note 5 for the derivation. $k_{\text{B}}$ is the Boltzmann constant and $T$ is the absolute temperature.

The free energy due to the mixing entropy of RBPs and solvent molecules has the form

$$\frac{f_{\text{mix}}}{k_{\text{B}}T} = \phi_p \log \phi_p + \left(1 - \phi_p - \left(1 + \alpha_p\right)\phi_r\right) \log\left(1 - \phi_p - \left(1 + \alpha_p\right)\phi_r\right).$$

(14)

The mixing free energy, Eq. (14), has the contributions of the mixing entropy of RBPs (the first term) and the mixing entropy of solvent molecules (the second term).

The free energy due to the interactions between RBPs has the form

$$\frac{f_{\text{int}}}{k_{\text{B}}T} = -\chi\left(\phi_p + \alpha_p \phi_r\right)^2,$$

(15)

where $\chi$ is the interaction parameter that accounts for the attractive interactions. For simplicity, we assumed that RBPs bound to the terminal regions of nascent pre-rRNAs are equivalent to RBPs freely diffusing in the DFC layer and that solvent molecules are equivalent to nascent RNA units in terms of the interactions.

The free energy due to the binding of RBPs to nascent RNAs has the form

$$\frac{f_{\text{bnd}}}{k_{\text{B}}T} = \phi_r\left[\alpha_p \log \alpha_p + \left(1 - \alpha_p\right) \log\left(1 - \alpha_p\right) + \epsilon \alpha_p\right],$$

(16)

where $\epsilon k_{\text{B}}T$ is the energy increase due to the binding of RBPs to nascent RNAs. For simplicity, we assumed that each nascent RNA unit has one binding site of RBPs.

**Free energy of system.** The free energy $F$ of the system, Eq. (2) is derived by using the fact that the number of microphases in the system is $3V_{\text{m}}/(4\pi r_{\text{in}}^3)$ to derive this form. The interfacial tension between GC and nucleosol is relative small[3,13] and FC is excluded out from the nucleolus by the transcription inhibition[3,14], implying that the interfacial tension between FC and nucleosol is even smaller. In vitro experiments suggest that the attractive interaction between FBLs is much larger than that between nucleophosmins, a major component of GC[13]. It implies that the interfacial tension between DFC and GC and that between DFC and FC are larger than their difference. By neglecting the difference between these interfacial tensions, the interfacial tensions $\gamma_{\text{in}}$ and $\gamma_{\text{ex}}$ at the interior and the exterior interfaces are represented by the forms

$$\gamma_{\text{in}} = \gamma_p\left(\phi_p\left(r_{\text{in}}\right) + \alpha_p\left(r_{\text{in}}\right)\phi_r\left(r_{\text{in}}\right)\right)$$

(17)

$$\gamma_{\text{ex}} = \gamma_p\left(\phi_p\left(r_{\text{ex}}\right) + \alpha_p\left(r_{\text{ex}}\right)\phi_r\left(r_{\text{ex}}\right)\right)$$

(18)

where $\gamma_p$ is the surface tension between a nucleosol and a liquid of the RBPs and is proportional to $\chi$.

**Lateral osmotic pressure.** The lateral osmotic pressure $\Pi_\parallel$ can be defined by

$$\Delta\Pi_\parallel = \sigma_{\text{in}}^2 \frac{\partial}{\partial \sigma_{\text{in}}} \left(\frac{F_{\text{d}}}{4\pi r_{\text{in}}^2 \sigma_{\text{in}}}\right),$$

(19)

where the derivative with respect to $\sigma_{\text{in}}$ is taken with the condition that $4\pi\sigma_{\text{in}} r_{\text{in}}^2$ is constant (this increases the area of the surface of a FC without changing the composition of the DFC layer). $\Delta\Pi_\parallel$ is the lateral osmotic pressure, from which the contribution of the isotropic osmotic pressure is subtracted.

**Radius of FCs at free-energy minimum.** The free energy $F$ is represented as a function only of the radius $r_{\text{in}}$ of FCs if the occupancy, $\alpha_p$, and the volume fractions, $\phi_p$ and $\phi_r$, are substituted into Eq. (2). The radius $r_{\text{in}}$ at the steady sate is derived by the condition $\frac{dF}{dr_{\text{in}}} = 0$ with the condition that $\sigma_{\text{in}} \propto r_{\text{in}}$ (this increases the area of the surfaces of FCs with the optimization of the arrangement of the active rDNA repeats). This leads to the form

$$\Delta\Pi_\parallel - \left(\gamma_{\text{in}} + \gamma_{\text{ex}} \frac{r_{\text{ex}}^2}{r_{\text{in}}^2}\right) + r_{\text{in}}\left(\frac{\partial \gamma_{\text{in}}}{\partial r_{\text{in}}} + \frac{\partial \gamma_{\text{ex}}}{\partial r_{\text{in}}} \frac{r_{\text{ex}}^2}{r_{\text{in}}^2}\right) = 0,$$

(20)

which represent the fact that the surface tension is balanced to the surface pressure generated by the FBRs of nascent pre-rRNAs.

**Cell culture, drug treatment, reverse transcription-quantitative PCR (RT-qPCR), immunofluorescence, and quantification of the size of FCs.** HeLa cells were maintained in DMEM containing high glucose (Nacalai Tesque, Cat# 08458-16) supplemented with 10% FBS (Sigma) and Penicillin-Streptomycin (Nacalai Tesque). Cells were treated with BMH-21 (Selleckchem, Cat# S7718) or CX-5461 (AdooQ Bioscience) for 3 h. Quantification of nascent pre-rRNAs (forward primer: 5'-CCTTCCCCAGGCGTCCCTCG-3', reverse primer: 5'-GGCAGCGCTACCATAACGGA-3')[14] by RT-qPCR was performed using LightCycler 480 II (Roche). These primers are located in the 5' external transcribed spacer region of pre-rRNA, where the mature 18S, 5.8S, and 28S rRNAs do not include this region. Therefore, these primers only detect pre-rRNAs and do not detect the mature rRNAs. GAPDH mRNAs were used as a loading control[17]. The antibodies to UBF (Santacruz, F-9, sc-13125, mouse monoclonal antibody, 1:50 dilution), FBL (Proteintech, 16021-1-AP, rabbit polyclonal antibody, 1:500 dilution), and NPM1/Nucleophosmin (Abcam, ab183340, SP236, rabbit polyclonal antibody, 1:100 dilution) were used to visualize FCs, DFCs, and GCs in immunofluorescence, respectively. Cells were grown on coverslips (Matsunami; 18 mm round) and fixed with 4% paraformaldehyde/PBS at room temperature for 10 min. Then, the cells were washed three times with 1x PBS, permeabilized with 0.5% Triton-X100/PBS at room temperature for 5 min, and washed three times with 1x PBS. The cells were incubated with 1x blocking solution (Roche, Blocking reagent, and TBST [1x TBS containing 0.1% Tween 20]) at room temperature for 1 h. Then, the coverslips were incubated with primary antibodies in 1x blocking solution at room temperature for 1 h, washed three times with TBST for 5 min, incubated with secondary antibodies (anti-mouse IgG, Alexa Fluor 488, superclonal [Thermo Fisher Scientific, Cat#A28175], anti-rabbit IgG, Alexa Fluor 568 [Thermo Fisher Scientific, Cat#A11036]) at room temperature for 1 h, and washed three times with TBST for 5 min. The coverslips were mounted with a Vectashiled hard-set mounting medium with DAPI (Vector, H-1500). Super-resolution images were acquired using ZEISS LSM900 with Airyscan 2. Quantification of the FCs marked by UBF staining within the nucleoli labeled by NPM1 staining was performed using NIS

Elements Advanced Research (NIKON). "FillArea" and "Max-Feret" of the UBF foci (FCs) detected with an intensity threshold were quantified as area and Lx of the FCs, see Fig. S4.

**Statistics and reproducibility**. Statistical analyses were performed using the Kruskal–Wallis test with Dunn's multiple comparison test for Fig. 7b, c, and Fig. S3b, c. The calculated p-values are described in the figure legends. No statistical method was used to pre-determine the sample size, but our sample sizes are similar to those reported in previous publications. All experiments were performed independently two or three times with similar results obtained. Prism 8 software (GraphPad) was used for the statistical analyses.

**Reporting summary**. Further information on research design is available in the Nature Portfolio Reporting Summary linked to this article.

## Data availability

The data of the numerical calculations available in figshare with the identier (https://doi.org/10.6084/m9.figshare.16599446)[58].

## Code availability

The Mathematica file used for the numerical calculations is available in figshare with the identifier (https://doi.org/10.6084/m9.figshare.16599446).

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

## Acknowledgements

This research was supported by KAKENHI grants from the Ministry of Education, Culture, Sports, Science, and Technology (MEXT) of Japan [to T. Yamamoto (20H05934, 21K03479, 21H00241), T. Yamazaki (21H00253, 22H02545), K.N. (19K06478, 22K06083), and T.H. (20H00448, 20H05377,21H05276)], J.S.T., PRESTO Grant Number JPMJPR18KA (to T. Yamamoto), the Mochida Memorial Foundation for Medical and Pharmaceutical Research (to T. Yamazaki), the Naito Foundation (to T. Yamazaki), the Takeda Science Foundation (to T. Yamazaki), and JST CREST Grant Number JPMJCR20E6 (to T.H.). T. Yamazaki thanks Toyofumi Kameoka (NIKON SOLUTIONS CO., LTD.) and Takayuki Funato (NIKON SOLUTIONS CO., LTD.) for support of microscopic image analyses. T. Yamamoto acknowledges the fruitful discussion with Noriko Saito (Japanese Foundation for Cancer Research), Yuma Ito (Tokyo Institute of Technology), Satoru Ide (National Institute of Genetics), Tsutomu Suzuki (Univ. of Tokyo), Takehiko Kobayashi (Univ. of Tokyo), Yutetsu Kuruma (JAMSTEC), Shintaro Iwasaki (Univ. of Tokyo), Sumio Sugano (Chiba Univ.), Haruhiko Siomi (Keio Univ.), Hideaki Matsubayashi (Tohoku Univ.), and Naomichi Takemata (Kyoto Univ.).

## Author contributions

T. Yamamoto, T. Yamazaki, K.K., and T.H. designed the research and wrote/revised the manuscript. T. Yamamoto developed the theoretical model and performed the calculations. T. Yamazaki performed the experiments.

## Competing interests

The authors declare no competing interests.
