## [Peer review file · Communications Biology]

Reviewers' comments:

Reviewer #1 (Remarks to the Author):

In this manuscript, the authors formulate a simple and effective mathematical formulation that captures the components and parameters which play a role in ribosomal RNA (rRNA) processing and maturation. rRNA transcription occurs within layers of phase-separated droplets called fibrillar centers (FCs), which are present within another layer called dense fibrillar component (DFC). Both these are part of a larger droplet called the granular compartment (GC). The model the authors present helps us understand how rRNA transcripts along with pol-1 suppress the growth of FCs. By carefully considering the factors that play a role in this process such as ribonucleoproteins, kinetic rates of transcription and chain elongation, partitioning into sub-phases etc., as well as many thermodynamic parameters such as free energies, osmotic pressure, and radii of the droplets, they present a model by which they predict the regions of nascent pre-rRNAs as responsible for osmotic pressure differences that depend on the radius of FCs and transcription rate.

This work is simple yet a well-designed study that models the complicated process of rRNA processing in the nucleolus. Their model is well supported by cell culture experiments. This work is important for biology in general but particularly for advancing the field of biomolecular condensates. This reviewer has no major criticisms except one:

In considering the parameters, the authors have not considered the uridylation and methylation of rRNA, which are important in their processing, which is thought to take place in DFC layer. Both these events significantly affect the charges and the chemistry of the molecule, which could impact its interfacial tension. The authors need to clarify this omission or include them in their calculations.

Reviewer #2 (Remarks to the Author):

Yamamoto and colleagues extensively modeled the different compartments of the nucleolus and have proposed a hypothesis to explain the structural make up of the nucleolus based on phase separation and the transcription of as well as the physical state (compressed versus stretched/RBP bound versus not) of nascent rRNA in which the rRNA acts as an arcRNA to organize the nucleolus as a whole. Previous work has demonstrated that the size of the FCs is increased when transcription of the rRNA genes is inhibited. Based on the modeling in this paper, one would expect that this is not an all or nothing phenomenon, but rather that the size of the FCs is scalable based on either the rate of Pol I transcription of the rRNA genes or possibly the absolute amount of pre-rRNA present in the nucleolus. This hypothesis holds true when the authors use sub-saturating amounts of their Pol I inhibitors, as the size of FCs scales with the amount of treatment used.

One control that would be extremely helpful for the author's conclusions would be to show that Pol I transcription is only partially inhibited under their sub-saturating conditions, possibly through in vitro transcription assays. If this result is known from previous work it should be stated.

It is appreciated that the authors included the methodology for measuring FCs in their supplemental materials. However, it is critical that they also show demonstrations of these measurements for the drug treated cells and not just the no drug case. As the fluorescence seems to be more diffuse in those samples in figures 7, S2 and S3 it is not entirely clear how the authors determined the boundaries of the FCs in the non-standard condition. The authors decisions on how to measure FCs could strongly influence the results.

Also, the authors need to address how their RT-qPCR assay only measures pre-rRNA and not fully processed mature rRNA in ribosomal subunits. Without critical information on this assay, it is possible that their RT-qPCR is measuring total ribosomal RNA amounts in their cells (based on the dearth of information in reference 11 Ide et al. 2020 about the method). If this is the case, figure 7 panels d and e are not accurate representations of the absolute amount of pre-rRNA and the results will be skewed as a result.

Finally, the authors will need to be careful to draw too strong of a conclusion that it is the rate of transcription that dictates FC size/localization and not simply the presence and amount of pre-rRNA in the particles that affects these things. Absent an experiment where authors can also increase rRNA transcription, it is hard to differentiate between the two possibilities.

Reviewer #3 (Remarks to the Author):

The study investigates the molecular reasons for the stable sizes of fibrillar centres in nucleolus. Specifically, the authors ask the question of why the FCs do not grow in size unless the transcription of pre-rRNA is inhibited. The authors develop a mean-field theoretical model to address this question considering many different factors including the dimensions of condensates, rate of elongation of rRNA, many interaction parameters, and interfacial effects. The work is of interest and suggests that the lateral pressure due to transcription suppresses condensate growth which is not present in the absence of active transcription. Finally, the authors show that suppression of transcription rate by

specific drugs enhances the dimensions of FCs in a dose-dependent manner. The theoretical model is presented well but I have several major concerns that the authors should address:

1) RNAs are inherently unstable. It is not clear to be why the authors haven't considered this in their model. Would RBP binding have to do anything with the apparent stability assumption?

2) It would be interesting to see a time-dependent plot of FC size as Pol1 transcription is continually suppressed (to mimic the effect of drugs in experiments). This could be performed under different combinations of chi and epsilon parameters. It would provide a much better picture of the model output, in my view.

3) Nucleolus and other condensates have tens of other molecules which could potentially play a role in maintaining the dimensions of FCs. Could the authors comment on it and explain why they rely primarily on Pol1 transcription rate to put forth their theory?

4) The dense nature of DFC and its invariant size could also originate from the stiffness of the RBP bound rRNAs (analogous to lipids as the authors suggest in discussion). Would this alone explain the apparent effect on the FC sizes?

5) A table describing each parameter (as there are many parameters used by the authors) and its value would be helpful at the beginning of the article. The current Table 1 does not have all the parameters named/labelled and it is tough go back and forth across supporting information, methods and the text.

Reply to Reviewer #1:

In this manuscript, the authors formulate a simple and effective mathematical formulation that captures the components and parameters which play a role in ribosomal RNA (rRNA) processing and maturation. rRNA transcription occurs within layers of phase-separated droplets called fibrillar centers (FCs), which are present within another layer called dense fibrillar component (DFC). Both these are part of a larger droplet called the granular compartment (GC). The model the authors present helps us understand how rRNA transcripts along with pol-1 suppress the growth of FCs. By carefully considering the factors that play a role in this process such as ribonucleoproteins, kinetic rates of transcription and chain elongation, partitioning into sub-phases etc., as well as many thermodynamic parameters such as free energies, osmotic pressure, and radii of the droplets, they present a model by which they predict the regions of nascent pre-rRNAs as responsible for osmotic pressure differences that depend on the radius of FCs and transcription rate.

This work is simple yet a well-designed study that models the complicated process of rRNA processing in the nucleolus. Their model is well supported by cell culture experiments. This work is important for biology in general but particularly for advancing the field of biomolecular condensates. This reviewer has no major criticisms except one:

Thank you very much for your interest and constructive comments. Your comment was useful to improve the discussion of our manuscript. We revised our manuscript in line with your comment. The revised parts were highlighted in red. Our reply to your comment follows:

1. In considering the parameters, the authors have not considered the uridylation and methylation of rRNA, which are important in their processing, which is thought to take place in DFC layer. Both these events significantly affect the charges and the chemistry of the molecule, which could impact its interfacial tension. The authors need to clarify this omission or include them in their calculations.

Thank you very much for this interesting comment. Our treatment is motivated by recent experiments showing that unprocessed long pre-rRNAs increase under fibrillarin depletion and it is rescued by either of the wild-type fibrillarin or the methyltransferase-null fibrillarin (Yao et al. Mol. Cell, 2019). The latter experiments also showed that the

intrinsically disordered region of fibrillarin is necessary to assemble the DFC layer, but the methylation of pre-rRNAs plays a relatively minor role. In general, the origin of the interfacial tension is the same as the origin of the assembly (S. A. Safran, *Statistical Thermodynamics of Surfaces, Interfaces, and Membranes*): molecules at interfaces are relatively unstable because they have fewer interacting partners than the interior of the condensate. We thus treated nascent pre-rRNAs as scaffolds and fibrillarin as the main driving force for the assembly of the DFC layer as well as the FC/DFC and FC/GC interfacial tensions in our minimal model. This does not exclude the possibility that other chemical modifications are involved in the processing of the terminal region of pre-rRNAs. We agree with the reviewer that the contribution of the chemical modification of RNAs to the assembly of nuclear bodies is an interesting problem and would like to return to it in our future study. Experiments to characterize the relationship between the size of FCs and DFCs and the extent of the chemical modification of pre-rRNAs would greatly help us to take into account the chemical modification in an extension of our model. Motivated by your comment, we added a sentence on L373-377:

It is of interest to extend the present theory by taking into account the co-transcriptional RNA processing to study the biochemical role of RBPs in the RNA processing⁴⁵, the modulation of RBP binding by the chemical modification of pre-rRNAs, such as methylation and pseudouridylation, and the role of the RNA chemical modification in the assembly of the multi-phase structure of nucleus.

Reply to Reviewer #2:

Yamamoto and colleagues extensively modeled the different compartments of the nucleolus and have proposed a hypothesis to explain the structural make up of the nucleolus based on phase separation and the transcription of as well as the physical state (compressed versus stretched/RBP bound versus not) of nascent rRNA in which the rRNA acts as an arcRNA to organize the nucleolus as a whole. Previous work has demonstrated that the size of the FCs is increased when transcription of the rRNA genes is inhibited. Based on the modeling in this paper, one would expect that this is not an all or nothing phenomenon, but rather that the size of the FCs is scalable based on either the rate of Pol I transcription of the rRNA genes or possibly the absolute amount of pre-rRNA present in the nucleolus. This hypothesis holds true when the authors use sub-saturating amounts of their Pol I inhibitors, as the size of FCs scales with the amount of

treatment used.

Thank you very much for your interest and constructive comments. Your comments were useful to improve the clarity of our manuscript and make the discussion deeper. We revised our manuscript in line with your comments. The revised parts were highlighted in red. Our point-by-point reply to your comments follows:

1. One control that would be extremely helpful for the author's conclusions would be to show that Pol I transcription is only partially inhibited under their sub-saturating conditions, possibly through in vitro transcription assays. If this result is known from previous work it should be stated.

Indeed, recent experiments have shown that the range of BMH-21 concentrations (< 0.5 μM) is well within the subsaturation regime (ref. (28)). We thus added a sentence in L301-302:

The range of concentrations of BMH-21 used in our experiments is well within the subsaturation regime of Pol I transcription (31).

2. It is appreciated that the authors included the methodology for measuring FCs in their supplemental materials. However, it is critical that they also show demonstrations of these measurements for the drug treated cells and not just the no drug case. As the fluorescence seems to be more diffuse in those samples in figures 7, S2 and S3 it is not entirely clear how the authors determined the boundaries of the FCs in the non-standard condition. The authors decisions on how to measure FCs could strongly influence the results.

As suggested by the reviewer, we have added the data on the detection of FCs during quantitative analysis under drug treatment conditions. To maintain the clarity of the figure, the data at the intermediate concentration of the BMH-21 condition (0.125 μM) are shown as representative data. We chose this BMH-21 treatment condition because, as shown in the quantification results in Figure 7, there are FCs of similar size in the other BMH-21-treated conditions (0.0625 and 0.25 μM), and thus we deemed it appropriate as a representative example. Although, as with most image quantification, we recognize the limitation that 100% accuracy cannot be achieved, the FCs appear to be accurately detected in many cases of BMH-21 treated cells. We believe that by quantifying a large number of FCs (n=500), we

were able to quantify the overall trend with sufficient accuracy.

3. Also, the authors need to address how their RT-qPCR assay only measures pre-rRNA and not fully processed mature rRNA in ribosomal subunits. Without critical information on this assay, it is possible that their RT-qPCR is measuring total ribosomal RNA amounts in their cells (based on the dearth of information in reference 11 Ide et al. 2020 about the method). If this is the case, figure 7 panels d and e are not accurate representations of the absolute amount of pre-rRNA and the results will be skewed as a result.

We used the following primers for pre-rRNA quantification:

Forward primer: 5'-CCTTCCCCAGGCGTCCCTCG-3'

Reverse primer: 5'-GGCAGCGCTACCATAACGGA-3'

These primers are located in the 5'ETS region of pre-rRNA, where the mature 18S, 5.8S, and 28S rRNAs do not include this region. Therefore, these primers only detect pre-rRNAs and do not detect the mature rRNAs.

4. Finally, the authors will need to be careful to draw too strong of a conclusion that it is the rate of transcription that dictates FC size/localization and not simply the presence and amount of pre-rRNA in the particles that affects these things. Absent an experiment where authors can also increase rRNA transcription, it is hard to differentiate between the two possibilities.

As you have noticed, in our theory, the osmotic pressure is generated by the fact that nascent pre-rRNAs are tethered to the surfaces of FCs and increases with the transcription rate because the surface density of nascent pre-rRNAs increases. Indeed, the kinetics of transcription itself can affect the conformation of nascent pre-rRNAs and thus affect the lateral pressure of the DFC layer. But, we estimate that its contribution is relatively small because the Pol I elongation is much slower than the relaxation of the terminal region of pre-rRNAs, see Supplementary Note 2. If the Pol I elongation is fast enough, we expect that nascent pre-rRNAs are condensed at the vicinity of the FC/DFC interface. We thus added sentences in L323-330:

The lateral pressure is generated by the fact that the nascent pre-rRNAs are tethered to

the surfaces of FCs via Pol I and attract fibrillarin to the DFC layer. This pressure increases with the transcription rate by increasing the surface density of the nascent pre-rRNAs. Thus, if the Pol I transcription is continuously and slowly suppressed, the size of FCs increases following eq. (4), see also Fig. 6. The contributions of the kinetics of transcription itself to the lateral pressure remain to be experimentally determined, while we estimate that its contribution is relatively small, see Supplementary Note 2.

We also added sentences that address that the relaxation dynamics of the terminal regions of nascent pre-rRNA is faster than the elongation L173-176:

We derive these quantities by analyzing the condition of the minimum of the free energy, Supplementary Note 1. The relaxation dynamics of the terminal region of nascent pre-rRNAs is estimated to be faster than the elongation and thus its contribution is negligible, see Supplementary Note 2.

Reply to Reviewer #3:

The study investigates the molecular reasons for the stable sizes of fibrillar centres in nucleolus. Specifically, the authors ask the question of why the FCs do not grow in size unless the transcription of pre-rRNA is inhibited. The authors develop a mean-field theoretical model to address this question considering many different factors including the dimensions of condensates, rate of elongation of rRNA, many interaction parameters, and interfacial effects. The work is of interest and suggests that the lateral pressure due to transcription suppresses condensate growth which is not present in the absence of active transcription. Finally, the authors show that suppression of transcription rate by specific drugs enhances the dimensions of FCs in a dose-dependent manner. The theoretical model is presented well but I have several major concerns that the authors should address:

1) RNAs are inherently unstable. It is not clear to be why the authors haven't considered this in their model. Would RBP binding have to do anything with the apparent stability assumption?

Our theory takes into account the RNA processing, with which the terminal regions of nascent pre-rRNAs are cleaved via the processing time τ_{pr} . The instability that can be

relevant but was not taken into account in our theory is thus the aberrant degradation of nascent pre-rRNAs before the cleavage of the terminal regions. Experimentally, full length pre-rRNAs have been observed under the depletion of the processing factors of pre-rRNAs (16). We thus believe that the aberrant degradation of pre-rRNAs is not significant. However, the latter experiments were performed on culture cells and it is possible that RBPs bound to pre-rRNAs may suppress the activity of endoribonuclease. We thus added sentences on L373-377:

It is of interest to extend the present theory by taking into account the co-transcriptional RNA processing to study the biochemical role of RBPs in the RNA processing⁴⁵, the modulation of RBP binding by the chemical modification of pre-rRNAs, such as methylation and pseudouridylation, and the role of the RNA chemical modification in the assembly of the multi-phase structure of nucleus.

2) It would be interesting to see a time-dependent plot of FC size as Pol I transcription is continually suppressed (to mimic the effect of drugs in experiments). This could be performed under different combinations of chi and epsilon parameters. It would provide a much better picture of the model output, in my view.

Motivated by your comments, we added a sentence in L323-328:

The lateral pressure is generated by the fact that the nascent pre-rRNAs are tethered to the surfaces of FCs via Pol I and attract fibrillarin to the DFC layer. This pressure increases with the transcription rate by increasing the surface density of the nascent pre-rRNAs. Thus, if the Pol I transcription is continuously and slowly suppressed, the size of FCs increases following eq. (4), see also Fig. 6.

The parameters, ϵ and χ , represent the binding energy of fibrillarin and the terminal region of pre-rRNAs and the magnitude of the multi-valent interaction between fibrillarin RNA-binding proteins, respectively. In principle, these parameters do not change with the suppression of Pol I transcription, but by the mutation of the intrinsically disordered regions (IDRs) and the RNA-binding regions of fibrillarin. To clarify how these parameters are relevant to experiments, we added sentence on L183-185:

In principle, these parameters, ϵ and χ , do not change with the suppression of Pol I transcription, but by the mutation of the RNA-binding regions and the intrinsically

disordered regions of fibrillarin, respectively.

3) Nucleolus and other condensates have tens of other molecules which could potentially play a role in maintaining the dimensions of FCs. Could the authors comment on it and explain why they rely primarily on Pol1 transcription rate to put forth their theory?

We used two types of RNA polymerase I inhibitors with different mechanisms of action, BMH-21 and CX-5461, and showed that treatment with these inhibitors causes FCs to fuse and eventually become a large single subcompartment (as also shown in Fig. 7 of our manuscript). In addition, previous works have shown that inhibition of RNA polymerase I by another RNA polymerase inhibitor (low dose of actinomycin D) or rapid degradation of RNA polymerase I by the AID degron system (Ide et al., Sci Adv (2020) 6, eabb5953) cause similar fusion of FCs. These data strongly suggest that RNA polymerase I transcription of pre-rRNAs is a major cause of this reorganization of FCs, and also suggest that other factors are not sufficient to maintain the multiphase structure, in which FCs are dispersed in a GC. However, we do not exclude the possibility that the other factors are involved in the size control of FCs. Also, more experimental inputs are necessary to take into account such molecular factors in the theory. Motivated by your comment, we added a sentence on L378-380:

Sixth, we neglect the potential contributions of molecular factors other than the RNP complexes of nascent pre-rRNAs and RBPs.

4) The dense nature of DFC and its invariant size could also originate from the stiffness of the RBP bound rRNAs (analogous to lipids as the authors suggest in discussion). Would this alone explain the apparent effect on the FC sizes?

Thank you for this very interesting comment. We expect that if the RBP bound terminal regions of pre-rRNAs are stiff, it will also explain the dependence of the FC size on the transcription rate because the lateral osmotic pressure in the DFC layer is mostly due to the translational entropy of fibrillarin localized in the layer. Motivated by your comment, we added a sentence on L377-378:

Fifth, we did not take into account the modulation of the stiffness of pre-rRNAs by the binding of RBPs.

5) A table describing each parameter (as there are many parameters used by the authors) and its value would be helpful at the beginning of the article. The current Table 1 does not have all the parameters named/labelled and it is tough go back and forth across supporting information, methods and the text.

Thank you for the great idea. We added a list of symbols as Supplementary Table 1 (because only 10 display items are allowed in the main article). We also added a sentence to notify the presence of this list in L94:

We here construct a minimal model of a nucleolus to predict the size of FCs in the steady state (the list of symbols is given in Supplementary Table 1).

REVIEWERS' COMMENTS:

Reviewer #1 (Remarks to the Author):

The authors have addressed my comments satisfactorily.

Reviewer #2 (Remarks to the Author):

In their resubmission, the authors have largely addressed my concerns from my first round of comments and I am largely satisfied with their response.

It is of vital importance that the method for FC quantification be transparent as they can be small compact regions under certain conditions shown and much more spread out in others. The authors have sufficiently addressed my concerns about their methodology here.

I appreciate the explanation of how their qRT-PCR assay detects pre-rRNA specifically in the response to reviewers. However, this information is also vital to the paper and should be in the methods section. Without this information, it is impossible to accurately understand the experiment being done.

While additional experiments that could either increase or decrease the rate of transcription by PolI by other means than direct inhibitors of the enzyme and the resulting measurements of FC size would be appreciated, this reviewer understands that it is likely outside the scope of this work as it is currently presented.

Reviewer #3 (Remarks to the Author):

The authors have satisfactorily addressed my comments. I recommend publication.

Reply to Reviewers comments:

Reply to Reviewer #1

The authors have addressed my comments satisfactorily.

Thank you very much for your recommendation. Our manuscript was greatly improved by your constructive comments.

Reply to Reviewer #2

In their resubmission, the authors have largely addressed my concerns from my first round of comments and I am largely satisfied with their response.

It is of vital importance that the method for FC quantification be transparent as they can be small compact regions under certain conditions shown and much more spread out in others. The authors have sufficiently addressed my concerns about their methodology here.

I appreciate the explanation of how their qRT-PCR assay detects pre-rRNA specifically in the response to reviewers. However, this information is also vital to the paper and should be in the methods section. Without this information, it is impossible to accurately understand the experiment being done.

Thank you for your additional constructive comments. We agree with you that the method for FC quantification must be transparent and that the explanation of our approach to specifically detect pre-rRNAs should be also in Method section. We thus added a sentence at L541:

These primers are located in the 5' external transcribed spacer region of pre-rRNA, where the mature 18S, 5.8S, and 28S rRNAs do not include this region. Therefore, these primers only detect pre-rRNAs and do not detect the mature rRNAs.

Revised parts are highlighted in red.

While additional experiments that could either increase or decrease the rate of transcription by Poll by other means than direct inhibitors of the enzyme and the resulting measurements of FC size would be appreciated, this reviewer understands that it is likely outside the scope of this work as it is currently presented.

Thank you very much for suggesting an additional experiment. It is an interesting experiment, but collaboration with experts of nucleolus is probably necessary to perform it. We hope that our manuscript motivates experts of nucleolus to perform additional test of our theory and acts as driving force of further experiments.

Reply to Reviewer #3

The authors have satisfactorily addressed my comments. I recommend publication.

Thank you very much for your recommendation. Our manuscript was greatly improved by your constructive comments.

REVIEWERS' COMMENTS:

Reviewer #1 (Remarks to the Author):

The authors have addressed my comments satisfactorily.

Reviewer #2 (Remarks to the Author):

In their resubmission, the authors have largely addressed my concerns from my first round of comments and I am largely satisfied with their response.

It is of vital importance that the method for FC quantification be transparent as they can be small compact regions under certain conditions shown and much more spread out in others. The authors have sufficiently addressed my concerns about their methodology here.

I appreciate the explanation of how their qRT-PCR assay detects pre-rRNA specifically in the response to reviewers. However, this information is also vital to the paper and should be in the methods section. Without this information, it is impossible to accurately understand the experiment being done.

While additional experiments that could either increase or decrease the rate of transcription by PolI by other means than direct inhibitors of the enzyme and the resulting measurements of FC size would be appreciated, this reviewer understands that it is likely outside the scope of this work as it is currently presented.

Reviewer #3 (Remarks to the Author):

The authors have satisfactorily addressed my comments. I recommend publication.